# MixtureKit: A General Framework for Composing, Training, and Visualizing Mixture-of-Experts Models

## Abstract

We introduce MixtureKit, a modular open-source framework for constructing, training, and analyzing Mixture-of-Experts (MoE) models from arbitrary pre-trained or fine-tuned models. MixtureKit currently supports three complementary methods: (i) *Traditional MoE*, which uses a single router per transformer block to select experts, (ii) *BTX* (Branch-Train-Mix), which introduces separate routers for each specified sub-layer enabling fine-grained token routing, and (iii) *BTS* (Branch-Train-Stitch), which keeps experts fully intact and introduces trainable stitch layers for controlled information exchange between hub and experts. MixtureKit automatically modifies the model configuration, patches decoder and causal LM classes, and saves a unified checkpoint ready for inference or fine-tuning. We further provide a visualization interface to inspect per-token routing decisions, expert weight distributions, and layer-wise contributions. Experiments with multilingual code-switched data (e.g. Arabic-Latin) show that a BTX-based model trained using MixtureKit can outperform baseline dense models on multiple benchmarks. We release MixtureKit as a practical foundation for research and development of MoE-based systems across diverse domains. The library is accessible at: *Link will be provided upon acceptance*.

## 1 Introduction

Building Large Language Models (LLMs) is an area of growing interest for researchers and practitioners within the AI community. This growth has been driven by the impressive performance of these models in a variety of downstream tasks (Yang et al., 2024) and has been accompanied by a rapid increase in the release of open-source models (Castaño et al., 2023). These models cover diverse domains, such as healthcare (Sellergren et al., 2025; Luo et al., 2022), software development (Rozière et al., 2024; Hui et al., 2024), and legal practice (Wu et al., 2023; Yang et al., 2023). To achieve higher performance and to expand the models' abilities in handling new tasks, previous work emphasized the need to scale up the models (Kaplan et al., 2020), resulting in large dense models with up to trillions of parameters (Team et al., 2025; Grattafiori et al., 2024). Despite the substantial value these models bring across domains and in users' daily tasks, they demand tremendous computational resources for optimal training and inference (Cottier et al., 2025). Furthermore, Luo et al. (2025) discovered that, during continual fine-tuning, catastrophic forgetting is more pronounced in larger models, likely due to their initial strong performance. On the other hand, Haque (2025) reported that smaller models mitigate catastrophic forgetting and retain learning capacity, although they exhibit lower overall performance.

To leverage the advantages of both scales, it became imperative to introduce a new architecture combining large model capacity with the ability to update only a subset of parameters, reducing future forgetting, and maintaining a lower inference budget. Mixture-of-Experts (MoE) models (Shazeer et al., 2017) efficiently scale LLM capabilities by routing different inputs to specific sub-networks, allowing for fewer parameter updates. With this new architecture, multiple models have been made available to the community that rank among the best published models, while offering outstanding performance and relatively low inference costs (DeepSeek-AI et al., 2025; OpenAI et al., 2025; Yang et al., 2025). However, the standard practice of pre-training MoE models from scratch incurs high computational costs. Moreover, this approach does not provide any control over the specific

domain of each expert, as it assumes that final convergence will occur after training without any prior knowledge of the domains (Sukhbaatar et al., 2024). Osborne (2024) Furthermore, the work by Osborne et al. (2024) highlighted that numerous models, including former state-of-the-art ones, remain unused after being surpassed by newer models. Therefore, it is necessary to recycle outdated *pre-trained* and *fine-tuned* models into a unified state-of-the-art model, thus reducing computational costs and providing a strong initialization for further fine-tuning.

In this work, we introduce `MixtureKit`, an open-source python library for advanced Mixture-of-Experts architectures that helps accomplish this requested recycling. Whereas some previous work has laid a solid ground for implementing certain architectures, it has primarily focused on well-known model families like Mistral, Llama, Phi, etc. Accordingly, previous efforts have not provided a generalization beyond these families that could also accommodate custom-code models built on specialized architectures. This package represents an additional effort to encourage the reuse of *pre-trained* or *fine-tuned* models, thereby substantially reducing computational training costs. In addition, it automates the creation of the new MoE-based model, allowing users to concentrate on the subsequent stages of the training workflow without requiring extra skills. Our main contributions are the following:

- **Unified composer class**: The package provides a one-line merge function and a complete training pipeline, enabling full MoE training and usage without extensive low-level expertise. It automatically adjusts the model configuration, applies patches to the decoder and causal LM classes, and generates a unified checkpoint ready for inference or fine-tuning. High-level approach-specific helper functions are integrated into the new MoE model's configuration, with necessary adjustments performed through regex pattern matching.

- **Advanced MoE strategies and load balancing**: The package supports advanced MoE schemes that focus on reusing HuggingFace checkpoints. These include *Traditional MoE*, *Branch-Train-miX* (BTX) and *Branch-Train-Stitch* (BTS). The implementations, previously unavailable to the open community, are now accessible with full support for all model families, enabling further research and experimentation. In addition, it provides an implementation of the load-balancing principle among experts, dealing with the known inactive specialists issue in MoE models (Fedus et al., 2022).

- **Token routing visualization and statistics**: To ease interpretation and provide deeper insights into the model's internal decision-making process, the package includes a visualization tool that traces token routing paths at two levels: (1) High-level visualization, where each token is colored based on the expert that received the most votes across transformer blocks, revealing dominant routing choices, and (2) Low-level visualization, where each token is quantified with expert-specific percentage, reflecting its precise contribution to the corresponding output either per layer or on average.

## 2 RELATED WORK

**Mixture-of-Experts (MoE)**    The work of Jacobs et al. (1991) introduced a new supervised learning paradigm for a system composed of multiple specialized neural networks, or experts coordinated by a gating network. As the gating network assigns inputs to different experts, each expert network consequently learns to specialize in a particular subset of the input space. They emphasized that this approach can achieve more efficient learning than a single dense network, especially for tasks subject to interference effects. By confining weight updates to a subset of parameters, the system mitigates catastrophic forgetting and avoids the slow learning that often occurs in large single-network models when changes in one part of the network disrupt unrelated tasks.

Shazeer et al. (2017) revived the Mixture-of-Experts principle with their seminal paper. The ability of the network to store and represent information is constrained by its number of parameters. *Conditional computation*, in which only specific parts of the network are activated for each input, has been theoretically proposed as a means of significantly increasing model capacity without corresponding increase in computational cost. They proposed a *Sparsely-Gated Mixture-of-Experts* (MoE) layer, where the gating network selects a sparse subset of $k$ experts to process each input, *top-k* gate-routing mechanism. The Switch Transformer (Fedus et al., 2022) and GLaM (Du et al., 2022), introduced by Google, were the pioneering models that demonstrated the effectiveness and scalability of the sparse MoE layers at the trillion-parameter level. The former replaced the feed-forward network

(FFN) layers with Switch layers containing multiple expert FFNs with a *top-1* gating mechanism, in which each token was routed to a single expert for processing. The study demonstrated that a Switch Transformer with 1.6 trillion parameters exhibited higher efficiency than smaller dense models, such as the 11 billion-parameter T5-XXL, when trained with equivalent computational resources. GLaM employed a more sophisticated gating mechanism, whereby each token was directed to the *top-2* experts from a total of 64 experts per MoE layer. Nevertheless, these two models were not released as open-weight models for further experimentation by the community.

Mixtral (Jiang et al., 2024) was the first open-weight model pre-trained from scratch using a mixture of 8 experts, where 2 are activated per input token. It has been shown to match or outperform LLaMA 2 70B and GPT-3.5 in standard benchmarks. Its sparse design enables inference approximately six times faster than dense counterparts, improving computational efficiency and practicality for real-world deployment.

**Model Merging**  The concept of model merging can be regarded as a related, albeit distinct, paradigm compared to traditional ensemble methods such as Random Forests (Chamma et al., 2023). It is evident that both approaches leverage the knowledge of multiple models; however, a distinction emerges in their execution. Ensemble methods integrate the predictions of independently trained models during inference, thus improving robustness but resulting in a substantial computational burden. In contrast, model merging techniques such as Model Soups (Wortsman et al., 2022) combine the parameters of fine-tuned models, often through simple averaging, into a single model that retains their collective strengths without incurring additional inference cost, marking a pivotal innovation that distills the advantages of ensembles into a deployable unified model. Inspired by *ensemble learning* and *parameter averaging*, Li et al. (2022) introduced a new principle, *Branch-Train-Merge* (BTM), in which a base model was replicated multiple times, each replica was trained on a separate subset of the original data, and the final output was obtained by combining the predictions of all replicas. As for inference, they allow for either merging all experts into a single model to improve efficiency or selectively utilizing only the most relevant experts for a specific task to maximize performance. Despite its strong performance, this architecture limited the ability to further fine-tune the individual experts' components within the unified structure.

**MergeKit (Goddard et al., 2025)**  `MergeKit`, delivered by Acree-AI, is an open-source library focusing on providing the community with easy access to various merging techniques through a YAML syntax configuration file. It enables the combination of model checkpoints, allowing their parameters to be merged to improve both performance and versatility, in response to the rapid growth of open-source language models. It also provides Mixture-of-Experts (MoE) merging techniques, where the implemented gates can be initialized under: *hidden* (based on the hidden state representations of the provided positive and negative prompts), *cheap_embed* (using only the raw token embeddings of the prompts) and *random* (randomly initializing the MoE gates). Nonetheless, the final model can only follow the direct MoE configuration of `Mixtral`, `DeepSeek` or `Qwen`, and therefore does not generalize. Moreover, the package does not include open-source implementations for newly developed MoE merging techniques.

**Mergoo**  `Mergoo`[1], delivered by Leeroo-AI, was the first package to deliver open-source implementations of new MoE-based merging techniques such as BTX (section 3.1), along with simplified configuration and walkthrough examples. However, this package focused mainly on hand-crafted model configuration files for known families such as Phi3, Mistral, and LLama, which limits its expansion to additional families.

## 3  MIXTUREKIT: DESIGN AND IMPLEMENTATION

`MixtureKit` provides a method-agnostic pipeline that composes a set of pretrained experts into a single checkpoint and patches the target architecture with either token routers (Traditional, BTX) or stitch layers (BTS). The system is driven entirely by a configuration dictionary (Fig. 2) and exposes a uniform interface that makes extending to new MoE variants a matter of registering a conversion rule for targeted submodules.

---

[1]https://github.com/Leeroo-AI/mergoo

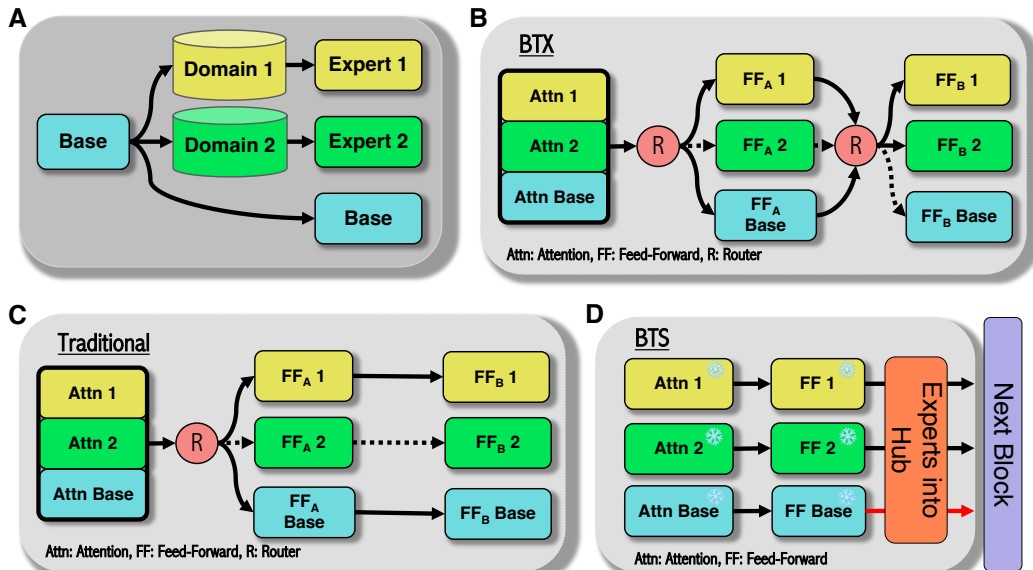

Figure 1: The workflow for building the new MoE-based unified model. (**A**) The common core component, where experts are prepared via continual pre-training or fine-tuning. This step encompasses the trained checkpoints hosted on the HuggingFace Hub (Wolf et al., 2020). A replica of the base model is also included to transfer knowledge to the new model. Figures (**B–C**) show *Branch-Train-miX (BTX)* and *Traditional MoE* routing strategies respectively. The straight lines represent the experts selected under the top-2 routing scheme, while the dashed line indicates the optional additional expert that may be chosen depending on the configured value of top-$k$. $FF_A$ and $FF_B$ denote the internal projection components within the Feed-Forward networks, such as gate, up and down. (**D**) Steering-based strategies such as *Branch-Train-Stitch (BTS)* where the input token is forwarded through both the frozen base and expert models. A trainable stitch layer is integrated to adjust the activations. The red line presents the final output derived from the base model after refinement with the experts.

## 3.1 IMPLEMENTED METHODS

**Branch-Train-miX (BTX)** Whereas the previous direction was to pre-train an MoE model from scratch without any prior assignment of domain specialties to experts, a process that demanded tremendous computational resources for efficient training (Mu & Lin, 2025), Sukhbaatar et al. (2024) introduced a recycling strategy called **Branch-Train-miX** (BTX) shown in Fig. 1-B. Following the work of (Li et al., 2022), they started from the same premise where multiple instances of the selected base model were trained on different partitions of the original dataset, as illustrated in Fig. 1-A. The feed-forward layers of these models are reconfigured as individual experts within a new MoE layer, while a trainable routing network directs each token to the most appropriate expert(s) according to the chosen mechanism. Unless otherwise specified, layers such as attention and embeddings are merged through parameter averaging across the instances, yielding a unified backbone. The unified MoE model can, in principle, be adapted using two fine-tuning strategies. An option is to update only the integrated routers while keeping all other parameters frozen, which offers computational efficiency but results in a slight degradation of performance; alternatively, full fine-tuning of all model components can be employed, particularly after parameter averaging, which is more computationally demanding but necessary to achieve optimal accuracy and fully realize the potential of the composed model. Despite promising results, this method relies on decomposing the components of different experts and reintegrating them according to the MoE framework; as a result, the merged model does not support the flexibility and interpretability offered by a modular structure, in which experts are kept separate and intact.

**Traditional MoE** As shown in Fig. 1-B, *BTX* introduces independent routers at each internal projection within the FFN (gate, up, and down projections) which provides a unified model with

greater degrees of freedom in selecting a specific path among experts. However, this comes at the cost of increased routing complexity. To align with the architecture presented in the work by Jiang et al. (2024), we propose a variant of *BTX* called **Traditional MoE** as presented in Fig. 1-C. which employs routing through entire FFN blocks. For every token, the router chooses one expert (or top-$K$ experts), and the selected experts are then used consistently across all internal projections. The unified model can be fine-tuned in the same way as BTX, but with fewer parameters since there are fewer routers. Although inputs are routed through the different experts, some of them are unlikely to be selected for processing, leading to the phenomenon of *dead experts*. The key to bypassing this issue is the addition of a loss, referred to as the load balancing loss, with a hyperparameter $\alpha$, which encourages the model to distribute workload more evenly among experts. However, a trade-off is needed in selecting $\alpha$ to prevent both significant degradation in the model's performance and instability during training.

**Branch-Train-Stitch (BTS)**  Although router-based models paved the way for the construction of recycled MoE models inspired by ensemble learning and parameter averaging, they lack a plug-in format that would allow for the seamless integration of new experts specialized in additional domains. To produce a unified model that is more inherently interpretable, Zhang et al. (2025) described a new flexible method called **Branch-Train-Stitch** (BTS). As presented in Fig. 1-D, once all experts (i.e. replicas) have been trained on the respective subsets of the original data, no parameter averaging is applied in this method; instead, each expert is retained as-is in the final model structure. To interchange information between experts and the hub (i.e. base) model, a special trainable bidirectional *StitchLayer* is added on top of selected transformer blocks, including the final block, at a predefined frequency to steer the corresponding activations. This layer consists of linear projections and a linear gate, with only one direction active at a time, while all the parameters of the underlying transformer blocks remain frozen. The first direction, `experts-into-hub`, refines the activations of the target transformer block. It linearly projects the hidden states of the experts in the hidden representation space of the hub model and computes the contributions of each expert, including the hub itself, to the output. This is done by first applying a linear gate ($w_{gate}$) to the hub's hidden state ($h_0$), followed by a softmax across all experts and the hub to obtain their relative contributions. The second direction, `hub-into-experts`, adjusts the activations of the experts by first linearly projecting the hub's hidden state into the hidden representation space of the experts. Instead of using a softmax function, this direction applies a sigmoid function to determine the relative contribution of the hub and each expert's previous hidden state to the updated states. BTS enables the merging of specialized experts in a flexible plug-in format, where training focuses mainly on the stitch lightweight layers, making it efficient and requiring low computational cost. However, one drawback of this approach is the memory challenge that may arise when fitting all the full experts

## 3.2 END-TO-END WORKFLOW

```
config = {
  "moe_method": "btx",
  "stitch_freq": 5,
  "model_type": "new_model_type",
  "num_experts_per_tok": 2,
  "experts": [
    {"expert_name": "base_expert", "model_id": "expert_base_checkpoint"},
    {"expert_name": "expert_1",    "model_id": "expert_1_checkpoint"},
    {"expert_name": "expert_2",    "model_id": "expert_2_checkpoint"},
  ],
  "router_layers": ["mlp.gate_proj", "mlp.up_proj", "mlp.down_proj"],
  "alpha": 0, "router_layers_index": []
}
```

Figure 2: Example of configuration dictionary `config` merging the feed-forward layers of three models under the Branch-Train-miX strategy (BTX) with two experts activated and load balancing disabled.

**User-Centric Configuration.** The entire build process is controlled by a single dictionary configuration object, shown in Fig. 2. The key fields directly specify the design choices of the unified MoE model. The `moe_method` selects the integration strategy (e.g., `btx`, `traditional` or `bts`), while `model_type` defines the identifier of the unified checkpoint. The list of `experts` provides the Hugging Face model IDs for the base model and the domain-specialized checkpoints for merging. Sparsity at inference time is controlled by `num_experts_per_tok`, which determines how many experts are activated for each token. This setting is equivalent to the top-$k$ parameter in routing, where only the $k$ highest-scoring experts are selected to contribute to the final output. The fields `router_layers` and `router_layers_index` specify which submodules (e.g., `mlp` for MoEs or `attn` for MoAs (Fu et al., 2024)) and which indexed transformer blocks are converted into MoE form, respectively. Additional parameters such as `stitch_freq` (for stitch-based methods) and `alpha` (for load-balancing regularization) further control the training behavior. Once defined, a single call to `build_moe(config)` produces a fully functional, `transformers`-compatible checkpoint. This makes experiments repeatable and easily shareable without requiring modifications to the model code.

**Expert Composition.** The process begins with the `compose()` method, where each expert model is loaded and its parameters are iteratively integrated into a unified state dictionary. For each parameter, the system determines whether it should be shared between experts or assigned individually to an expert-specific namespace. The shared parameters are averaged across all experts with shape-aware alignment, ensuring compatibility even when the hidden dimensions differ slightly. Parameters designated for expert-specific conversion are stored under structured namespaces (e.g., `experts.expert_i.weight`), producing a coherent MoE parameter layout that remains fully compatible with the `transformers` library.

**Architecture Patching and Method Specialization.** After expert composition, `MixtureKit` rewrites the destination model's architecture to host the MoE extensions. This process is orchestrated in the `save_checkpoint()` function, which copies the original `modeling_<base>.py` and `configuration_<base>.py` files, renames them to match the new `model_type`, and updates imports and class names via `_replace_type_dependency`. Targeted edits then inject the MoE logic: `_replace_script` rewrites `nn.Linear` and `Conv1D` layers inside the `Attention` and `MLP` classes, while `_modify_decoder` and `_modify_model` adjust the decoder and model classes to incorporate gating and load-balancing losses.

For router-based methods (*Traditional*, *BTX*), the patched classes are used to replace the original linear layers with MoE-aware modules (`convert_linear_to_moe`), enabling a gating network that selects the top-$k$ experts at each step. In contrast, stitch-based methods (*BTS*) do not modify linear layers; instead, `_patch_stitches` augments the model's forward path by introducing parallel expert streams and inserting `StitchLayer` modules that blend hub and expert activations at configurable depths controlled by the `stitch_freq` parameter in the user configuration (Fig. 2). Importantly, these specializations change only the forward computation and parameter organization, while the composition and saving pipeline remain identical. This modularity ensures that the framework is naturally extensible to future MoE strategies with minimal additional code.

**Extensibility.** Adding a new MoE variant only requires implementing a small adapter module that provides a conversion function for the chosen layers or a stitch integration function for hub–expert fusion. Once registered in the configuration, the new method automatically benefits from `MixtureKit`'s loading, saving, and compatibility pipeline, without touching the core composer logic.

### 3.3 Visualization and Interpretability

An important aspect of `MixtureKit` is its visualization interface, which provides researchers with real-time information on token routing decisions. The tool is implemented as an interactive *Streamlit* application and supports all MoE configurations that rely on token-level routing, namely the *Traditional* and *BTX* variants.

**Routing Mechanism.** At the core of both Traditional and BTX routing is a gating function that maps each token representation to expert scores. Let $h_t^{(\ell,p)} \in \mathbb{R}^d$ denote the hidden state of token $t$

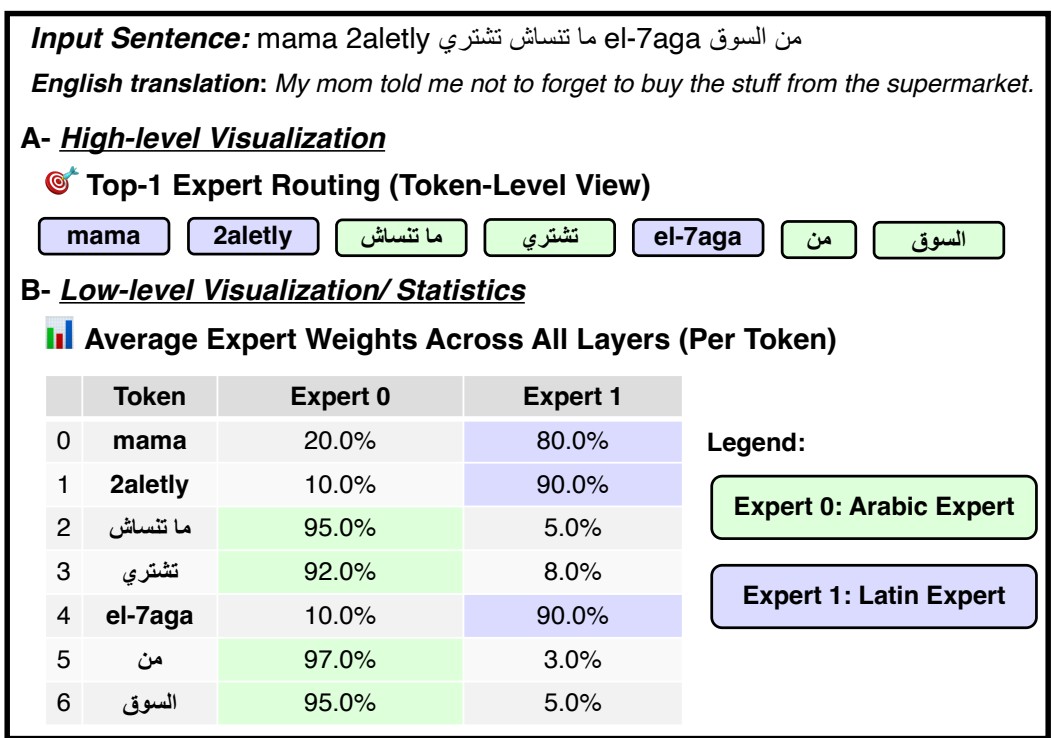

Figure 3: MixtureKit Token-Routing Visualizer, showing the main interface with the input prompt: (**A**) High-level color-coded token assignments showing expert specialization at a glance, and (**B**) Average expert weights across all selected layers for each token.

at transformer block $\ell$ and internal projection $p \in \{\text{gate}, \text{up}, \text{down}\}$. Each projection is paired with a gating matrix $W^{(\ell,p)} \in \mathbb{R}^{d \times E}$ that produces unnormalized logits with $E$ denoting the number of experts:

$$g_{t,e}^{(\ell,p)} = h_t^{(\ell,p)} W^{(\ell,p)}, \qquad e \in \{1, \dots, E\}.$$

The router then selects the top-$k$ experts

$$S_t^{(\ell,p)} = \text{TopK}\big(g_t^{(\ell,p)}, k\big),$$

and assigns normalized weights using a softmax function

$$w_{t,e}^{(\ell,p)} = \frac{\exp(g_{t,e}^{(\ell,p)})}{\sum_{j \in S_t^{(\ell,p)}} \exp(g_{t,j}^{(\ell,p)})}, \qquad e \in S_t^{(\ell,p)}.$$

In *BTX*, each projection (gate, up, down) has its own router $W^{(\ell,p)}$, so a token may follow different experts' components within the same FFN block. In *Traditional MoE*, a single router $W^{(\ell)}$ is shared across all three projections, ensuring a consistent expert choice throughout the block.

To provide an interpretable perspective on routing, MixtureKit's visualization aggregates and displays expert normalized weights across the full depth of the model. For token $t$, the aggregated contribution of expert $e$ is

$$\overline{w}_{t,e} = \frac{1}{|\mathcal{L}_t|} \sum_{(\ell,p) \in \mathcal{L}_t} w_{t,e}^{(\ell,p)}, \qquad \mathcal{L}_t = \{(\ell, p) : t \text{ is processed at block } \ell, \text{ projection } p\}.$$

Figure 3 illustrates both perspectives: (A) high-level token assignments based on the dominant expert $\arg\max_e \overline{w}_{t,e}$, and (B) detailed statistics showing the aggregated weights $\overline{w}_{t,e}$ throughout all transformer blocks and projections. This combination enables both intuitive inspection of token-level specialization and fine-grained analysis of how expert usage evolves across the model depth.

In addition, the MixtureKit visualization interface allows users to restrict the aggregation to specific projections (e.g., only gate or up layers) at specific transformer blocks, facilitating a more detailed examination of how expert choices evolve throughout the model.

This tool has proven valuable for diagnosing expert under-utilization, detecting routing collapse (where one expert receives most tokens), and studying expert specialization in code-switched or multilingual scenarios. It can be applied to any method where the contribution of each expert is easily accessible within the model.

# 4 PRACTICAL EXAMPLE: SRIPT-SPECIALIZED EXPERTS

Egyptian Arabic, also referred to as *Masri*, is the most prevalent Arabic dialect, with a population of over 100 million native speakers in Egypt and a high degree of mutual intelligibility throughout the Arab world. It exhibits notable disparities from *Modern Standard Arabic* (MSA) with respect to phonology, vocabulary, and grammar. A distinctive aspect of this dialect is its frequent use of two writing systems. Egyptian Arabic speakers often employ both the traditional Arabic script and a Latin-based script, commonly known as Arabizi or Franco-Arabic (Elnagar et al., 2021). Multi-script Egyptian Arabic can be addressed by integrating script-specific experts, one trained on Arabic-script data and another on Latin-script (Arabizi) data, into a unified Mixture-of-Experts (MoE) model that dynamically routes tokens to the appropriate expert. This modular design facilitates scalable adaptation across scripts while preserving high performance and efficiency.

To demonstrate the potential of the proposed library, we adopt the experimental setup from (Shang et al., 2025) and try to reproduce the results using `MixtureKit`. To begin with, the base model `Gemma3-4B-pt` [2] was continually pre-trained separately on available Arabic and Latin script datasets to develop script-specific experts. Secondly, the pre-trained experts were integrated with the base model employing the *BTX*, leading to a novel MoE model comprising three experts, two of which are active per input, with a total of 6B activated parameters. This configuration defines the announced *Nile-Chat-3x4B-A6B model* where incorporating the base model as an additional expert provided broader general knowledge and enhanced English capabilities, extending beyond the script-specialized experts. For comparison, the two script-specialized experts were also merged independently of the base model, resulting in the outlined *Nile-Chat-2x4B-A6B* variant. The unified MoE models were trained in two stages. Initially, Supervised Fine-Tuning (SFT) was performed employing a Low Rank Adaptation (LoRA) configuration with an alpha value set to $512$, a learning rate of $1e-4$, and an effective batch size of $256$. In order to maintain the efficacy of the English-centric base model, which functions as a third expert, a proportion of English instructions have been incorporated from Wild-Chat [3]. Secondly, Direct Preference Optimization (DPO) has been implemented as the final alignment stage.

| Model | Average Arabic | Average Latin | Translation Long (chrf) | Translation Short (chrf) | Transliteration (chrf) |
|---|---|---|---|---|---|
| **Nile-Chat-4B-Arabic-Expert** | 53.21 | 44.63 | 58.81 | 52.7 | 26.21 |
| **Nile-Chat-4B-Latin-Expert** | 48.85 | 48.06 | 37.09 | 31.27 | 80.59 |
| **Nile-Chat-4B** | 53.01 | 49.07 | 58.4 | 52.01 | 80.44 |
| **Nile-Chat-2x4B-A6B** | 55.87 | 52.32 | 61.59 | 53.71 | 83.89 |
| **Nile-Chat-3x4B-A6B** | 55.74 | 51.23 | **61.9** | **55.37** | **83.97** |
| **Nile-Chat-12B** | **60.0** | **52.61** | 60.61 | 53.53 | 80.97 |

Table 1: Performance comparison of Arabic/Latin experts, Nile-Chat dense models (4B and 12B) and *BTX*-based counterparts across Arabic, Latin and generation benchmarks. The highest scores are indicated in **bold**, the second-highest are underlined.

To evaluate the performance of the different experts and models, we followed the evaluation procedure described in previous work (Shang et al., 2025), focusing on the average of all Egyptian bench-

---

[2]https://huggingface.co/google/gemma-3-4b-pt
[3]https://huggingface.co/datasets/allenai/WildChat-1M

marks in both Arabic and Latin scripts (measured with *accuracy* or *normalized accuracy*), as well as the generation benchmarks related to translation and transliteration (measure with *chrF*). As illustrated in Table 1, we successfully reproduced the findings, showing that **Nile-Chat-3x4B-A6B** and **Nile-Chat-2x4B-A6B** offer a compromise between the dense 4B and 12B models for discriminative tasks in Arabic script ($53.01 < \underline{55.87} < \mathbf{60.0}$), perform comparably on Latin script ($\underline{52.32} \approx \mathbf{52.61}$), and outperform them in tasks requiring extensive text generation, achieving the top-2 highest scores across all translation and transliteration tasks and metrics.

## 5  CONCLUSION

In this paper, we introduced `MixtureKit`, providing a practical step forward to make Mixture-of-Experts (MoE) research widely accessible. By allowing the reuse of existing *pre-trained* or fine-tuned checkpoints, systematically composing them into unified models, and adapting architectures for various routing and stitching strategies, it reduces the need to train costly MoE models from scratch. Through its visualization interface, `MixtureKit` also makes expert behavior interpretable, helping diagnose routing imbalance and study specialization dynamics in multilingual or domain-specific settings. Beyond the strong empirical results we report, the modular design of the framework reduces the barrier to experimentation with new MoE variants, offering the community a flexible foundation for future research and development. The provision of an open-source toolkit for the merging of model checkpoints is intended to promote collaboration among researchers, developers, and practitioners worldwide, thereby fostering innovation and knowledge sharing.

## 6  FUTURE WORK

Since the library modifies submodules with MoE components based on regex pattern matching, it relies on the HuggingFace versioning of models' configuration files. We aim to support as many versions as possible. In addition, the new configuration files currently lack the optimizations introduced in recent inference packages, such as the recently introduced MoE kernels in *vLLM* [4]. MixtureKit is a dynamic project dedicated to continuously incorporating new methodologies through collaboration with the open-source community. Additionally, while most of the literature focuses on merging models with the same architecture, an interesting direction is to explore cross-architecture merging.

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

## A LLM USAGE

We use LLMs solely to polish writing and clarify ideas, keeping all scientific reasoning human-driven. The model acts only as a stylistic assistant, enhancing readability without contributing content.

