# OpenReview forum: "MixtureKit: A General Framework for Composing, Training, and Visualizing Mixture-of-Experts Models"
_ICLR.cc/2026/Conference — Submitted to ICLR 2026_

### Official Review · Reviewer_fZkE · 2025-10-29

**Soundness:** 2
**Presentation:** 3
**Contribution:** 2
**Rating:** 2
**Confidence:** 4

**Summary:**

This work presents MixtureKit, an open-source (not open-sourced yet, and authors claim that they will release the code upon acceptance) framework for constructing, training, and analyzing Mixture-of-Experts (MoE) models from arbitrary pre-trained or fine-tuned models. This codebase support not only traditional MoE but also some advanced versions like BTX and BTS.

**Strengths:**

1) MoE is widely used in most frontier open-sourced LLMs. It is important to have good codebase/infra to support MoE training and inference.
2) This codebase seems easy to use. Based on the examples shown in the submission, it seems easy and fast to setup the experiments and visualizations.

**Weaknesses:**

1) I'm not very sure about how useful it is to support the BTX adn BTS model architectures. To my best knowledge, there are very few frontier models using these designs.
2) There seems no further research-focused efficiency optimization for the MoE models in this codebase, although author also mentioned the expensive cost of MoE models is the pain point.
3) I understand training MoE and doing visualization is good. But there are many codebases can to some extent achieve this already.
4) It is not clear how to integrate this codebase with other LLM training codebases easily. For example, if we are using megatron to train an MoE model, how can we use this codebase's feature?
5) If this codebase can not be combined with existing codebase easily, how is the training/inference efficiency of the MoE implementation. Can it beat megatron or some other MoE-based LLM training frameworks?

**Questions:**

see above

---

> ### Author Response · Authors · 2025-11-19
>
> We thank the reviewer for the detailed comments they have provided. We will provide concrete answers that help to better highlight the goal of the paper.
> 1. We agree that BTX and BTS are not yet adopted in many frontier models. However, we see MixtureKit’s support for them as important for two reasons:
> * They are state-of-the-art recycling methods for turning multiple experts into a unified model, and currently lack widely usable, generic implementations. By providing open, HF-agnostic implementations, MixtureKit lowers the barrier for the community to experiment with them and potentially adopt them in larger models.
> * Our case study shows that BTX can already be competitive with dense baselines in a realistic multi-script scenario, suggesting that such methods are not just of theoretical interest.
>
> We will add a sentence in the introduction explicitly motivating why exposing BTX/BTS in a generic toolkit is valuable even if frontier models have not yet converged on these designs.
>
> 2.  MixtureKit is composed of two main parts: Merging and Training. Merging is scalable with introduced experts (we have tried merging 10 experts-replicas of Qwen3-4B) which depends on the loading and unloading of the corresponding expert to align the layers. Training is introduced at different levels: router-only, lora-based and full-training. In the experiment presented in the paper, we performed a lora-based training with a full-training for the router where the compute footprint was comparable with its dense counterpart while holding SOTA performance on some benchmarks.
>
> 3. The two parts that compose MixtureKit are independent. For the merging part, previous work focused on performing MoE merged models based on known families (such as Mixtral, DeepSeek, etc.) where a modeling (configuration) file is already supported and hosted on HuggingFace. MixtureKit goes beyond to support any family of models (even personalized models with special configurations) as long as they follow HuggingFace’s scheme. As for the training, this can be done either by the supported trainer in MixtureKit (that presents the ability of freezing all parameters except for router ones, which has a low compute footprint) or by any other training framework (We will add extra examples that show how the final model can be trained with other frameworks)
>
> 4. As mentioned, the scope of MixtureKit is not to compete with other training frameworks but to present a complementary phase for MoE-based model building with a special initialization from pretrained and fine-tuned models. One of the limitations declared in the paper is the need to validate its support for optimized fused kernels presented in the frameworks that can push further its training and inference processes.

---

> > ### Comment · Reviewer_fZkE · 2025-11-27
> >
> > Thanks author for the reply.
> > I decide to keep my score after reading.

---

### Official Review · Reviewer_npR5 · 2025-10-30

**Soundness:** 2
**Presentation:** 1
**Contribution:** 1
**Rating:** 2
**Confidence:** 5

**Summary:**

The paper **"MixtureKit: A General Framework for Composing, Training, and Visualizing Mixture-of-Experts Models"** introduces an open-source framework that enables users to easily build, combine, and interpret Mixture-of-Experts (MoE) architectures. Instead of pretraining large MoE models from scratch, MixtureKit allows the reuse of pretrained or fine-tuned models (experts) and merges them into a unified model through three methods: **Traditional MoE**, **Branch-Train-miX (BTX)**, and **Branch-Train-Stitch (BTS)**. The system automatically patches model configurations, integrates load balancing and gating mechanisms, and offers a visualization interface for analyzing token routing and expert utilization.

A practical case study demonstrates MixtureKit’s efficiency on **Egyptian Arabic**, combining Arabic- and Latin-script experts into a multilingual MoE that achieves competitive or superior results to dense baselines. The framework proves flexible, modular, and extensible, supporting future MoE strategies and integration with inference systems like vLLM.

**Strengths:**

**Originality**

The paper presents an open-source toolkit, **MixtureKit**, for composing and training Mixture-of-Experts (MoE) models from existing pretrained or fine-tuned checkpoints.


**Quality**
- The implementation appears **technically sound and modular**, providing a clear merging and patching pipeline compatible with the HuggingFace ecosystem.
- The inclusion of a **visualization interface** for analyzing token routing and expert utilization adds usability value.

However, there is no benchmarking against **state-of-the-art MoE systems** (e.g., Mixtral, DeepSeekMoE, Gemini-MoE), making it difficult to assess the real quality of the approach.

**Clarity**
- The writing is **easy to follow**, with clear explanations of each supported method and step-by-step workflow examples.
- Figures and configuration examples improve readability, and the paper succeeds in documenting how MixtureKit works in practice.


**Significance**
- As an **engineering contribution**, the framework may be useful to practitioners who wish to experiment with MoE model recycling or visualization.


**Overall Assessment:**
The paper demonstrates solid engineering effort and contributes a usable open-source tool for MoE experimentation. Yet, it offers little in terms of conceptual or empirical advancement. Its contribution aligns more with a *software release paper* rather than the type of **theoretical or methodological innovation** typically expected at ICLR.

**Weaknesses:**

1. **The motivation lacks depth and practical justification.**
   The paper positions MixtureKit as a “general framework for composing and training MoE models,” but fails to explain why such a framework is necessary beyond existing open-source systems (e.g., DeepSpeed-MoE, FairScale, Megatron-LM). There is no demonstrated bottleneck or pain point that MixtureKit specifically solves. As a result, the motivation appears weak and insufficiently grounded in practical or scientific needs.

2. **The contribution is primarily engineering integration rather than scientific innovation.**
   MixtureKit mainly combines existing MoE routing variants (Traditional, BTX, BTS) into a unified API and configuration structure. While the integration is convenient, it does not propose any new algorithmic insight, theoretical formulation, or training paradigm. The framework reuses well-established components (routers, experts, stitching layers) without extending their capabilities, leading to minimal conceptual increment.

3. **Experimental validation is extremely limited and unconvincing.**
   The evaluation focuses only on small-scale multilingual code-switching tasks (e.g., Arabic-Latin), without quantitative comparisons to stronger baselines such as Mixtral, DeepSeekMoE, or Switch Transformers. No large-scale or cross-domain results are reported, and there is no measurement of computational efficiency, scalability, or convergence stability. Consequently, the claims of generality and performance improvement are not empirically substantiated.

4. **The framework introduces non-trivial engineering cost with limited benefit.**
   The process of merging multiple pretrained experts into an MoE model requires substantial training and parameter alignment overhead. The paper does not discuss the additional compute cost or compatibility issues (e.g., tokenizer mismatch, architecture heterogeneity). Given the small gains shown in experiments, the practical benefit of using MixtureKit over existing fine-tuning methods appears marginal.

5. **Lack of comparison with existing model-merging and modular training frameworks.**
   Several recent frameworks—such as AdapterFusion, LoRAHub, and ModelSoup—already support model composition and knowledge integration. The paper fails to situate MixtureKit within this broader context, nor does it show advantages in efficiency, extensibility, or interpretability compared to these systems. Without such positioning, its originality and contribution to the field remain unclear.

6. **Evaluation metrics and visualization claims are superficial.**
   Although the paper highlights MixtureKit’s visualization interface, the provided analyses are descriptive rather than analytical. There are no quantitative insights into routing entropy, expert utilization balance, or load distribution variance. The visualization results serve as demonstrations, not as scientific evidence supporting the framework’s design.

**Questions:**

> I encourage the authors to thoroughly address the weaknesses and questions raised in this review. If the authors can provide detailed explanations and in-depth clarifications during the rebuttal, and if the revised version demonstrates substantial progress in both clarity and improvement, I will be willing to reassess the manuscript and **adjust my overall rating** accordingly, based on the quality and depth of the revision.

---

1. **On the motivation and positioning of MixtureKit:**
   - The paper claims to address a gap in the accessibility and flexibility of MoE research, but it remains unclear what concrete limitation in current MoE toolchains (e.g., DeepSpeed-MoE, Megatron-LM, Tutel, FairScale) MixtureKit resolves.
   - Could the authors provide a clear comparison table or discussion showing (a) which functionalities are missing in prior systems, and (b) how MixtureKit explicitly fills those gaps?
   - Is there any specific use case (e.g., domain adaptation, cross-lingual MoE composition, or model interpretability) where MixtureKit enables something *previously impossible or significantly easier*?
   - Without such clarification, the contribution risks appearing as an engineering duplication rather than a research advancement.

2. **On the claimed novelty and framework design:**
   - The integration of Traditional, BTX, and BTS routing strategies is technically convenient, but conceptually derivative. Are there any new routing mechanisms or training procedures introduced?
   - How does MixtureKit ensure stable optimization when combining heterogeneous experts (with different data distributions or pretraining domains)?
   - Are there novel algorithmic elements—such as adaptive router regularization, inter-expert consistency constraints, or new stitching layer formulations—that go beyond existing MoE literature?
   - If not, could the authors justify why this framework constitutes a scientific contribution rather than a software engineering release?

3. **On experimental design and empirical adequacy:**
   - The experiments focus solely on small multilingual code-switched datasets (Arabic–Latin). Why were these chosen as the main testbed, and how do they generalize to other domains such as programming or medical text?
   - The paper lacks a systematic analysis of model behavior (e.g., expert utilization rates, token routing distribution, or convergence stability). Could the authors include quantitative visualizations of these dynamics?
   - Can MixtureKit reproduce or surpass established MoE baselines like Mixtral, DeepSeekMoE, or Switch Transformers under equivalent compute budgets?
   - Without strong empirical grounding, how should readers interpret the claimed generality and performance improvements?

4. **On computational cost, scalability, and resource implications:**
   - Merging multiple pretrained experts into a single MoE model can introduce parameter redundancy and routing overhead. What is the actual compute footprint (FLOPs, GPU-hours, or memory consumption) compared to dense or LoRA-based fine-tuning?
   - How does MixtureKit scale when combining 10+ experts or larger backbones (e.g., 13B–70B models)? Are there known performance bottlenecks in the current implementation (e.g., routing latency, checkpoint merging time)?
   - Does the framework support partial expert loading or lazy routing to reduce inference cost?
   - Have the authors evaluated how the mixture composition affects downstream latency or deployment efficiency?

5. **On comparison with other model-merging and modular adaptation frameworks:**
   - Several recent systems—such as AdapterFusion, ModelSoup, LoRAHub, and MergeKit—already provide mechanisms for checkpoint merging and modular fine-tuning.
   - Could the authors clarify how MixtureKit differs in architecture abstraction, efficiency, and extensibility?
   - In particular, what advantages does MixtureKit offer in cases involving *heterogeneous* model architectures or tokenizers, where existing systems already struggle?
   - A formal benchmarking comparison or ablation (e.g., merging time, performance gain per parameter) would strengthen the claim of contribution.

6. **On visualization and interpretability of expert behavior:**
   - The visualization interface is presented as a major feature, yet the paper only provides qualitative screenshots rather than analytical evidence.
   - Could the authors provide quantitative interpretability metrics—such as routing entropy, expert sparsity ratio, or per-layer specialization index—to substantiate the claim that MixtureKit helps diagnose model behavior?
   - Are these visual tools integrated into the training pipeline (e.g., for adaptive routing adjustments), or are they purely post-hoc inspection utilities?
   - How reproducible and generalizable are these visualization outputs across models of different architectures or token vocabularies?

7. **On future development and community impact:**
   - The authors mention ongoing support for more model versions and open-source collaboration. What governance or maintenance plan exists to ensure MixtureKit remains compatible with rapidly evolving LLM ecosystems (e.g., Llama 3, Mistral, Gemma)?
   - How can other researchers extend or plug in new router designs or expert composition methods into MixtureKit without modifying the source code?
   - Finally, does MixtureKit aim to serve primarily as a **research tool (for probing MoE behavior)** or as a **production framework (for model deployment and merging)**? The intended scope is currently ambiguous.

---

> ### Author Response · Authors · 2025-11-19
>
> We thank the reviewer for the detailed comments they have provided. We will provide concrete answers that help to better highlight the goal of the paper.
>
> 1- a) Existing MoE toolchains largely fall into two categories:
> * Training frameworks (DeepSpeed-MoE, Megatron-LM, FairScale, Tutel): these focus on training MoE models from scratch or with MoE layers already present in the architecture, often with custom CUDA kernels and specific supported backbones. They generally do not provide a principled way to (i) merge arbitrary pre-existing HF checkpoints into a single MoE model, or (ii) modify HF modeling files automatically to add advanced routing schemes and visualizations.
> * Merging libraries such as MergeKit and Mergoo: these provide parameter averaging and limited MoE merging but are tightly coupled to specific model families (e.g., Mixtral, DeepSeek, Qwen) and do not expose advanced recycling methods like BTX/BTS as family-agnostic building blocks.
>
> MixtureKit is explicitly designed to fill this gap:
> * It operates at the HF checkpoint level, taking arbitrary pre-trained or fine-tuned models (including custom architectures) and composing them into a unified MoE checkpoint.
> * It then automatically patches the modeling and config files of the target architecture to add router or stitch layers without the user hand-editing model code.
> * It exposes this capability via a single configuration dictionary and a one-line ```build_moe(config)``` call, rather than requiring manual code forks per model family.
> This combination – automated checkpoint composition + architecture patching + method-agnostic configuration – is, to our knowledge, not provided by existing toolchains.
>
> 1- b) As mentioned, the main goal is to create a new unified checkpoint based on a special initialization from already pretrained or fine-tuned checkpoints (possible SOTA in selected domains). Previous cited works have partially focused on reducing this gap by the means of replicating the modeling file for a few known families such as Mistral, Llama, etc. MixtureKit goes beyond this limitation where it configures the modeling part of the model to widen the scope of accessibility for the community. It is also provided with a visualization tool that helps to interpret the inner decision mechanism of the created model.
>
> 2- a) As for the first version, MixtureKit is not introducing any new routing or training procedures. The initial goal was to provide an open-implementation for published methods such as BTX and BTS. The current status paves the way for future research and contributions with an additional inner-monitoring to highlight the stability of the training process.
>
> 2- b) When combining heterogeneous experts, there is a need to retrain the model on a few optimization steps from the original (or similar data) of the experts to homogenize the different components of the unified framework.
>
> 2- c) We consider MixtureKit as a combination of engineering and scientific contribution. From the engineering discussed parts, it helps the community create new models from existing checkpoints without the need for any additional changes based on the modeling file of the original base models. For the scientific part, this package prepares the full infrastructure for this merging part that finalizes with the new checkpoint. Later on, the supported trainer (with the ability to focus only on the added layers such as Stitch or Router) or any other trainers can be used. The only current limitation is the support for the MoE kernels that should be the focus in the future releases.
>
> 3- a) MixtureKit was developed during an initial project where the goal was to train a cross-script LM that is able to deal with both Arabic and Latin scripts (for Egyptian as instance). Following, instead of limiting the scope to train a dense model with all the available processed data and monitor the final result, we believed after the published work by Meta, that a special initialization of the inner-layers with the experts can help the model achieve a possible SOTA performance. MixtureKit was developed in the agnostic-model-family spirit (to support personalized models) and was tested within the same project where we validated the claims as described in the paper.
>
> 3- b) We thank the reviewer for highlighting this important point and we agree that these metrics are needed to measure training dynamics. They can be added in the next release of the package. In a revised version, we will add:
> * Routing entropy per layer,
> * Expert utilization histograms (fraction of tokens per expert), and
> * A simple specialization index (e.g., difference between expert usage on Arabic vs Latin tokens),

---

> > ### Author Response · Authors · 2025-11-19
> >
> > 3- c) This is also an important question raised by the reviewer. The initial aim of the project was to compare the final MoE checkpoint if it is on a par with its dense counterparts (which our experiments showed its performance to reach the level between 4B and 12B while outperforming under some tasks). In the next version, we can work on adding a comparison with established MoE such as Mixtral to empirically prove its efficiency.
> >
> > 3- d) Our current approach was composed on a mixture of validated claims by the original papers and an empirical validation proposed by the cross-script trained model. While the experiments showed a current SOTA performance on some benchmarks (based on the fact that the dense models are the actual SOTA models for these benchmarks), we are aware that the final model needs further tuning to reach its SOTA capacity (data recipe, load balancing loss, contrastive loss among experts, etc.).
> >
> > 4- a) As discussed in the paper, the package introduces flexible training procedures with only-router, lora-based or full-training are supported. We agree that additional compute footprint can better guide the reader on the utility of the package (it was discussed implicitly in the text). To achieve the full potential of the model, a full-training is required if possible. In the published experiments, we performed lora-based training (because of a lack of resources) while full-training the router modules. This experiment showed that the compute footprint for the MoE checkpoint is comparable to that of dense models (Continually pretraining and fine-tuning dense model vs Continually pretraining experts + SFT for the unified checkpoint).
> >
> > 4- b) We have tried merging up to 10 experts (10 replicas of Qwen3-4B). The merging is highly dependent on the loading process of the different experts (each expert is loaded and deloaded sequentially) which is fast and can be parallelized if needed. No bottlenecks have been observed in the merging procedure nor the training.
> >
> > 4- c) Partial expert loading and lazy routing are not part of the actual status of the package. We thank the reviewer for highlighting these principles as they can reduce the inference cost along with specialized fused kernels.
> >
> > 4- d) We have tried merging two cross-script experts (Arabic & Latin) vs an additional expert (base-model). We have observed that the former lacks the knowledge of the base model (when asked for English prompts, it was replying in Arabic or Latin as no knowledge has been transferred through merging).
> >
> > 5- a) We thank the reviewer for their question about the utility of MixtureKit. As aforementioned and declared in the paper, MixtureKit focuses on the merging techniques applied for MoE-based models without any prior conditional or limitation for the family of the model. We will make the comparison to AdapterFusion, ModelSoup, LoRAHub, MergeKit, etc., more explicit:
> > * AdapterFusion/LoRAHub operate at the adapter/LoRA level, not at the level of creating a new MoE architecture with token-level routing.
> > * ModelSoup/standard merging focus on parameter averaging across variants of the same model but do not introduce routers or experts.
> > * MergeKit and Mergoo are the closest in spirit, but as the paper notes, they either:
> >  – restrict MoE merging to specific families like Mixtral/DeepSeek/Qwen, or
> >  – rely on hand-crafted modeling files for a few model types, limiting generalization to new architectures.
> >
> > By contrast, MixtureKit:
> > * Automatically patches modeling files for arbitrary HF models,
> > * Provides open implementations of BTX and BTS
> > * Adds a visualization layer for routing behavior.
> >
> > We will add a short table summarizing these differences.
> >
> > 5- b) As mentioned in the future work, cross-architecture is still an interest for the authors to focus on in the next releases. The first version highlighted the use case of merging common-architectures to share the knowledge with the final model. However, we do agree that cross-architecture merging is an ongoing important direction of research that could bring more benefits.
> >
> > 6- a) The initial goal of the visualization tool was to measure the generation capacity of the model after being trained (which could highlight the router's ability to guide tokens into the right experts’ capacity). Further quantitative metrics can be easily integrated in the next release of the package with a focus on training-level metrics’ visualization for all available architectures.
> >
> > 6- b) The current implemented visualization tool can be considered as an ad-hoc inspection that examines the inner-decision making process of the model after being trained under the provided prompts. We do agree with the reviewer that training-based visualization (Router-entropy, Router-token distribution, etc.) are actually needed and can be easily integrated in the package.

---

> > > ### Author Response · Authors · 2025-11-19
> > >
> > > 6- c) The visualization tool is built in the same spirit of MixtureKit being agnostic (no prior condition on the architecture). The only dependency lies in the existence of the router layers that are further inspected to monitor the performance of the final checkpoint.
> > >
> > > 7- Finally, we clarify the intended role and maintenance of MixtureKit:
> > >
> > > a) Governance / maintenance: MixtureKit closely tracks the HF ecosystem: the composer works by transforming modeling files based on HF naming conventions, and we are committed to updating it as new families (Llama 3, Mistral, Gemma, etc.) appear. As of today, the package has been tested with the latest LLMs from different providers (known and unknown families).
> > >
> > > b) Extensibility: Researchers can add new router designs or expert-composition methods by implementing a small adapter module and registering it; they do not need to modify the composer core. We will add an explicit “how to add a new method” example to the documentation.
> > >
> > > c) Scope: MixtureKit is primarily a research tool; to probe MoE behavior, recycle checkpoints, and experiment with new routing schemes, but produces standard HF checkpoints that can then be integrated into production pipelines (e.g., deployed with vLLM or further fine-tuned with DeepSpeed/Megatron). We will clarify this dual role in the conclusion.

---

### Official Review · Reviewer_W5GP · 2025-11-04

**Soundness:** 2
**Presentation:** 2
**Contribution:** 3
**Rating:** 4
**Confidence:** 3

**Summary:**

This paper introduces a toolkit that allows researchers to use pre-trained Mixture-of-Expert (MoE) models and combine them together for stronger systems. Overall, this seems to be an interesting tool that might help people in the community. The paper has a good background and related works section. It is an engineering heavy paper that has limited experimentation – though would be perfectly appropriate as a system description or demo paper. There are significant novel advancements that this paper makes, but they are less empirical and more engineering (which is not necessarily a negative).

**Strengths:**

This appears to be a potentially very useful toolkit. Being able to use pre-trained experts and combine them in a much broader way than the current literature is very appealing.

**Weaknesses:**

I am not sure that the proposed toolkit actually accomplishes what it says. A bit more experimentation could be useful. For instance, in Table 1 the authors try to reproduce another paper using their method, but it is unclear to me whether or not the actually ever do. It would be nice to see this done, and perhaps one more paper as well using a different method and part of their toolkit to show it actually does everything they claim. This is not a paper that needs to beat SOTA results, but only replicate them - so experiments should be straightforward to run if the toolkit is as easy as they claim.

**Questions:**

This might be lack of awareness on my part, but are many MoE methods actually “experts” in a specific domain, or is it more of an abstraction where we do not know how a particular expert is getting a vector routed to it? For instance, in Figure 3, how common is it to have an actual Arabizi vs Arabic script expert? You mention routing collapse, but would that be a problem for something in different scripts like this?

In table 1, you try to replicate a paper and show a lot of results. However, it is unclear to me if you actually do replicate it. Do you? Or if not, how close are they? I’m not sure what row is your implementation and which row is the paper you are trying to reproduce.

---

> ### Author Response · Authors · 2025-11-19
>
> We thank the reviewer for the detailed comments they have provided. We will deliver concrete answers that help to better highlight the goal of the paper.
> 1. MixtureKit provides the ability to maintain a merged checkpoint with special experts from preselected domains. In MoE models pre-trained from scratch (e.g., Switch, Mixtral), experts are indeed discovered by default in an unsupervised way and are not annotated with explicit domains. MixtureKit is designed to support both:
> * Unsupervised experts: any set of checkpoints can be treated as experts, regardless of whether the scope include one of the preselected domains.
> * Explicit domain experts: in our case study, each expert is deliberately specialized: one expert is continually pre-trained on Arabic-script Egyptian data, another on Latin (Arabizi), and a base/hub model is trained more broadly. This delivers a true example of including embedded experts such as “Arabic expert” vs “Latin expert”.
>
> These experts are pretrained (or continually pretrained) to prioritize the selected domain (such as Arabic and Arabizi). Once merged, the initial status of the router is random, thus it doesn’t have any prior knowledge about the token-level routing pattern to the right experts. Thus, extra training (a few steps that don't have a big impact on the overall footprint) is needed. It is also worth mentioning that MixtureKit was built to create a cross-script model that could compete with its dense counterpart (which was validated in the experiment).
>
> Regarding routing collapse in this setting: we found that script-specialized experts are naturally well-separated in token space, and we further encourage balanced usage via the standard load-balancing loss. The visualization interface (Fig. 3) then confirms that Arabic-script tokens are routed predominantly to the Arabic expert and Latin tokens to the Latin expert, rather than collapsing into a single dominant expert. We will add a short paragraph clarifying that MixtureKit does not require domain-labeled experts, but supports them as a particularly interpretable use case.
>
> 2. In table 1, we do not replicate another paper but we do provide the actual scores that we have got with the MoE model (built with MixtureKit) achieved in the previous work. This experiment highlights the fact that MoE’s performance (at 6B activated) lies between 4B and 12B (further tuning can uncover its full potential) with an outstanding performance among generative tasks. This experiment highlights the utility of BTX (after publishing its open implementation in MixtureKit) to serve as an advanced MoE merging technique achieving remarkable scores.

---

### Official Review · Reviewer_8cDM · 2025-11-05

**Soundness:** 2
**Presentation:** 3
**Contribution:** 3
**Rating:** 4
**Confidence:** 3

**Summary:**

Mixture of Expert Models (MoEs) have shown significant promise, especially in recent years as it seems many large industry research organizations have moved towards MoE architectures to maximize total parameters while keeping active parameters (and therefore compute budgets) low. However, MoEs are complex to implement, generally requiring significant engineering investment. Though modern MoE LLMs have been regularly studied in larger research organizations, there is yet to be a single unified open-source library enabling independent researchers to investigate the range of MoE architectures with ease.
The primary contribution of this paper is a software library dedicated to training of Mixture of Experts (MoE) models with a variety of architecture options (vanilla MoE, BTX, BTS, etc) and additional visualization features to aid in analysis. The software enables easy usage of checkpoints pretrained from different libraries, merged as experts into a single model. The authors additionally include an example with experiments that they conduct in their framework, which comprise an implementation of Arabic and Latin script experts.

**Strengths:**

The difficulty of training MoE models has been a longstanding problem, and there is indeed a need for a library to simplify the MoE research process. The software, as described by the authors, seems to have a fairly simple and intuitive interface, created to be compatible with the HuggingFace library of models, and includes useful routing visualizations to support research.

**Weaknesses:**

In general, it's difficult to assess the strength of a software contribution. As a reviewer, it's impossible to know whether the implementation is truly as usable or intuitive as the authors describe. In line 381, the authors say "[t]his tool has proven valuable for…" but we do not receive any details about these scenarios where the tool has proven valuable
While the paper includes one set of experiments from the authors, it lacks any other point of comparison to support its implementation quality. An example of such a comparison would be a reimplementation of a subset of the exact experiments in the original BTX paper, BTS paper, or other notable papers using similar methods, demonstrating the MixtureKit reimplementation's evaluation results, as well as the configs, and any other code a user would need to write in MixtureKit to generate this reimplementation. As the paper is right now,, a reviewer can't assess whether MixtureKit is truly usable.
The authors call the implementation in S4 a "reimplementation." However, their results are exactly those of the paper they claim to reimplement, and it seems as though this is not a reimplementation from scratch, but rather the original implementation, or an implementation heavily informed by insider knowledge from the original authors, and thus does not qualify as evidence of the usability and accuracy of MixtureKit.
There is also a general lack of information about details in usage. Though the authors include one example config, there is no comprehensive list of available options.
The authors also say that "[a]dding a new MoE variant only requires implementing a small adapter module" but do not include an example file.

**Questions:**

1. Is Section 4 truly a reimplementation, from scratch, entirely separate from the original implementation of the paper?
2. Can you provide examples of usage? E.g., using your framework to do a true reimplementation of an existing paper, and an example of extension to new MoE variants?
3. How different of an MoE variant can be easily included? I.e. what assumptions does your framework make, which would actually be very difficult to override?

---

> ### Author Response · Authors · 2025-11-19
>
> We thank the reviewer for the detailed comments they have provided. We will deliver concrete answers that help to better highlight the goal of the paper.
> 1. MixtureKit was built during a previous project that aimed to build cross-script dense models, for instance Arabic and Arabizi in the Egyptian Language. To validate the performance of MoE merged models from specialized experts, MixtureKit was used to create a new checkpoint involving Arabic and Arabizi experts along with the base model (to perform knowledge transfer into the final checkpoint). In section 4, we highlighted the fact that these scores have been achieved in a previous work by using the term (reimplemented). We thank the reviewer for raising this issue as we suggest rephrasing it in a better way after acceptance that could describe it as a previous work. In the next version of the paper, we can perform additional experiments that can guide the reader to understand the high-utility of the package.
> 2. MixtureKit’s actual version supports the open implementations of two promising designs published by Meta (BTX and BTS) along with a special support for the vanilla routing mechanism. It is built according to an extensible paradigm that welcomes new routing techniques or MoE merge models contributed by the community. An additional spreadsheet for contribution and extensibility will be added to the package that better router the researchers and practitioners.
> 3. The core part of the package handles preparing the right configuration, mirrored by a one-line merge ability for the user. Two limitations of the actual status of the package have been described in the paper: One is related to support of fused optimized kernels that can accelerate the training (and permit the use of Expert Parallelism). Another limitation is related to assuming a HuggingFace Transformer architecture with identifiable submodules (MLP and Attention layers) that can be matched by regex. These are the only places where we patch in router or stitch modules. What is currently not yet supported is true cross-architecture merging (e.g., mixing models with different layer stacks or radically different tokenizers) and highly non-Transformer variants. We also flag cross-architecture merging as future work in the paper.

---

### Meta-Review · Area_Chair_rYmK · 2026-01-08

**Summary:**

Reviewers find this primarily an engineering contribution without sufficient novel research insights. Advantages over existing MoE toolkits are not demonstrated, and experimental validation is limited. Authors clarified design goals and use cases but did not provide additional experiments to validate that the toolkit reproduces claimed capabilities. The paper lacks the research depth expected for publication. I recommend rejection.

**Reviewer Concerns:**

see above

**Reviewer Scores:**

Discussion was sufficient; authors provided clarifications but no new experiments to address validation concerns, and scores would have remained similar.

---

### Decision · Program_Chairs · 2026-01-26

Reject